# Effect of Degradation in Small Intestinal Fluids on Mechanical Properties of Polycaprolactone and Poly-l-lactide-*co*-caprolactone

**DOI:** 10.3390/polym15132964

**Published:** 2023-07-06

**Authors:** Sam Peerlinck, Marc Miserez, Dominiek Reynaerts, Benjamin Gorissen

**Affiliations:** 1Department of Mechanical Engineering, KU Leuven, 3001 Leuven, Belgium; 2Flanders Make, 3001 Leuven, Belgium; 3University Hospital Gasthuisberg and Department of Development and Regeneration, KU Leuven, 3000 Leuven, Belgium

**Keywords:** polycaprolactone, poly-l-lactide-*co*-caprolactone, mechanical properties, hydrolytic, in vitro degradation, intestinal fluids

## Abstract

Polycaprolactone and poly-l-lactide-*co*-caprolactone are promising degradable biomaterials for many medical applications. Their mechanical properties, especially a low elastic modulus, make them particularly interesting for implantable devices and scaffolds that target soft tissues like the small intestine. However, the specific environment and mechanical loading in the intestinal lumen pose harsh boundary conditions on the design of these devices, and little is known about the degradation of those mechanical properties in small intestinal fluids. Here, we perform tensile tests on injection molded samples of both polymers during in vitro degradation of up to 70 days in human intestinal fluids. We report on yield stress, Young’s modulus, elongation at break and viscoelastic parameters describing both materials at regular time steps during the degradation. These characteristics are bench-marked against degradation studies of the same materials in other media. As a result, we offer time dependent mechanical properties that can be readily used for the development of medical devices that operate in the small intestine.

## 1. Introduction

Polycaprolactone (PCL) has gained traction as a biomaterial and is currently used in a plethora of applications, ranging from drug-delivery systems to implants and tissue scaffolds [1]. Characterized by a degradation time in the order of months, a low tensile strength and Young’s modulus [2], it has been discarded as a valid biomaterial for high load-bearing applications like plates, screws, etc. However, for applications that require structural compliance, PCL is a compelling material. This is especially true for medical devices or tissue engineering scaffolds that interface with softer tissue [3], making PCL a potential candidate for use in the hollow organs of the gastrointestinal (GI) tract. There, it can be used as an implantable tissue engineering scaffold [4,5] or in small intestinal stents that nowadays use more rigid materials [6,7]. To further expand the use of PCL in applications where even lower elastic moduli are desired, copolymers with l-Lactide chains can be integrated in the polymer network. The resulting polymer, poly-l-lactide-*co*-caprolactone, exhibits tunable properties, ranging from weak and waxy to tough and elastic [8]. Both the mechanical properties and degradation behaviour of PLCL are greatly influenced by the copolymer ratio of the l-Lactide to Caprolactone. poly-l-lactide-*co*-caprolactone with a copolymer ratio of 70 wt% l-Lactide and 30 wt% Caprolactone has gained special interests from the scientific community as it behaves rubber-like with an elastic modulus in the order of 10–100 MPa and ultimate strain of over 400% [9]. As such, Polycaprolactone and poly-l-lactide-*co*-caprolactone (70%:30%) (referred to as PLCL in this manuscript) is well suited for the development of implantable medical devices and scaffolds that target the intestine, which exhibit analogue properties [10]. While both PCL [2,11,12,13,14,15] and PLCL [9] are well described in literature in terms of synthesis processes and non-degraded mechanical properties, the degradation behavior in a small intestinal environment remains unknown to us.

The mechanism behind the hydrolytic degradation of PCL, that is expected to occur in the small intestine, are well reported [3], indicating a slow degradation rate that can be expedited by thermal, pH-mediated and enzymatic effects. Previous studies show that enzymatic activity speeds up the degradation of PCL films, sponges [16] and meshes, which were tested in both in vitro and in vivo environments [17]. More specifically, an in vitro [18] and in vivo [19] study of the weight loss of PCL films reported a faster decrease in weight in gastrointestinal fluids compared to in a phosphate buffer solution (PBS). PLCL is expected to experience similar degradation mechanisms to PCL [8], but specific degradation rates of its mechanical properties are not as elaborately reported. PLCL (with different copolymer ratios (40%:60%, 60%:40% and 80%:20%)) has been shown to exhibit vastly different degradation rates of its mechanical properties in PBS [20], indicating the importance of the exact copolymer ratio when investigating degradation, while subcutaneously implanted scaffolds of PLCL (with copolymer ratio 50%:50%) have shown slow degradation rates [21].

It is thus clear that the degradation of both PCL and PLCL have been shown to be sensitive to the specific environment they reside in; however, the specific progression of mechanical properties in the small intestinal environment is unknown. This knowledge is paramount for the design of PCL/PLCL medical devices that operate in the small intestine, especially when they are mechanically loaded. An incomplete knowledge of the material parameters throughout a device’s lifetime, could lead to premature failure. To avoid this, and with the help of finite element analysis (FEA), the entire lifetime of such devices can be simulated, however, to accurately predict their behavior in situ, accurate material properties are required. This study aims to provide a comprehensive and practically useful overview of the mechanical properties of Polycaprolactone (PCL) and poly-l-lactide-*co*-caprolactone with a copolymer ratio of 70%:30% (PLCL) and their degradation in human small intestinal fluids. To this end, PCL and PLCL dogbone-shaped tensile testing samples are fabricated using injection molding, submerged in collected human small intestinal fluids (at 37 °C and refreshed every day) for up to 70 days and subjected to uniaxial tensile testing which enabled the extraction of the degradation time-dependent properties of yield stress, Young’s modulus, elongation at break and the viscoelastic coefficients of both materials. The next section gives an overview of the procedures worked out for the treatment and testing of the materials, followed by a more elaborate discussion of the test results and a discussion.

## 2. Materials and Methods

### 2.1. General Procedure

To determine the progression of elastic and viscoelastic tensile properties of PCL and PLCL, samples are submerged in small intestinal fluids for over a period of up to 70 days. The tensile testing samples are dog bone shaped, and are divided in degradation groups, with 5 samples of each material in each group. The degradation groups are distributed by submersion time: 0, 1, 2, 3, 7, 14, 21, 28, 35, 42, 56, and 70 days, in coordance with previous degradation studies [8,18]. In total, 60 samples of each material are required for tensile testing, as destructive tests are envisioned. After their respective degradation periods, instantaneous elastic modulus (E), yield stress ($\sigma_{y}$), elongation at break and the viscoelastic relaxation Prony coefficients of each sample are measured and derived. Figure 1 schematically depicts the different methods used in this study, which are detailed in the paragraphs below.

### 2.2. Dogbone Sample Preparation

Medical ngrade PCL and PLCL are procured from Corbion (*Purasorb* PC08 and PLC7015 for PCL and PLCL respectively) in pellet form, and injection molded in a hardened steel dogbone-shaped die, using a FANUC Roboshot S-2000i 30B injection molding machine (FANUC Benelux, Mechelen, Belgium). The samples’ reduced section is nominally 16 mm long, 1.6 mm wide and 1 mm thick, with the aspect ratio of 10 eliminating significant boundary effects while stretching the sample. For a detailed drawing of the mould, we refer the reader to the Appendix A. PLCL pellets are first dried at 100 °C for 6 h as per suppliers instruction. PCL and PLCL are both injected at a low rate, with a holding pressure of 300 bar and low mould temperature of 30 °C. The barrel temperature is set at 180 °C for PCL and 200 °C for PLCL. After demolding, PCL samples show no sign of shrinking, while PLCL samples shrunk significantly, showing an increased section of 1.76 mm by 1.08 mm and reduced length of 13.8 mm.

### 2.3. Degradation Environment

The small intestinal fluids that are used as a degradation medium, are freshly collected from patients with (post operative) intestinal probes, stomas and as effluent from enterocutaneous fistulas from various patients. By doing so, we are able to mimic intestinal conditions that are representative for pathologies that warrant a possible use in the future for PCL and/or PLCL implants. Care is taken to mix fluids from various donors to limit inter-patient variability. After collection of the small intestinal fluids, MES-buffer (pH 6.5, Sigma-Aldrich M3671, Sigma-Aldrich, Saint Louis, MO, USA) at 25 m Molar and Antibiotic-Antimycotic (Life Sciences 15240062, ThermoFisher Scientific, Waltham, MA, USA) at 3 vol% are added for stability and to prevent growth of bacteria and fungi. As Figure 1C schematically depicts, this volume of degradation medium is then refrigerated and weekly replaced with freshly collected and treated medium. A tailor-made automated system distributes the refrigerated medium to a set of incubation tubes, each containing a specific degradation group of dogbone samples, which are kept at 37 °C. The fluids inside the incubation tubes are refreshed every 24 h. More details on the automated refreshing and distribution system are given in Appendix A. After submerging each group of samples for a dedicated time in the degradation medium, the samples are rinsed, labeled and photographed, before measuring their thickness using an in-house-developed Micro Laser Scanner built around a gocator 2010 line profile sensor (LMI Technologies Inc., Burnaby, BC, Canada). This procedure relies on the reflection of a line laser to generate a spatially discrete point cloud of the thickness of the measured sample. As this measurement proved unreliable for transparent PLCL, those samples were measured using vernier calipers. The samples were then finally subjected to the tensile testing protocol.

### 2.4. Tensile Testing

After degradation and characterizing the unloaded dimensions, the samples are subjected to uniaxial tensile testing on a (ZwickRoell, Ulm, Germany) biaxial tensile testing machine, as depicted in Figure 1D and described in detail in Appendix A. As the force measurement is conducted along the direction of the samples, forces perpendicular to the horizontal plane (i.e., buoyancy) are negligible. The samples are clamped using tailor-made 3D-printed Polylactic Acid (PLA) extension arms, enabling them to remain submerged in a saline solution at 37 °C during the test. The test was conducted under displacement control, measuring reaction forces. Further, the actual displacements are measured using the on-machine linear encoders and by optically tracking two ink markers placed on the reduced section of each sample (recorded by a VIC-Snap (correlated SOLUTIONS) camera with 1128 × 902 px resolution). The displacement profile imposed on the samples is schematically depicted in Figure 1D, according to the following protocol, which was adapted to the used tensile testing machines and samples [22,23]. After the sample is clamped and submerged, the test is initiated. Two linear ramp pull-hold-release cycles are then imposed, with each an individual preload setpoint, with the holding periods applied to test the viscoelastic properties. The amplitude of each hold displacement is a 4% and 5% increase of the preload displacement for PCL and PLCL samples, respectively, with strain rates of 0.2 and 0.3 mm/s, respectively. The tensile testing protocol is concluded by a pull until either failure or end-of-stroke at the same strain rate. For each sample, all data was recorded at a sample rate of 20 Hz.

### 2.5. Post-Processing

To derive the mechanical properties of the samples, different post-processing steps are performed. First, MATLAB image processing tools are employed to determine the planar strains occurring in the homogeneous cross section of the sample. Axial strain was measured based on the distance between sample marking, while for the reduction of sample width, the contrast between sample and background was employed. Second, these measurements are synchronized to the data from the linear encoder and load cells, as slight timing errors between the two systems occurred. Third, the true stress in the sample’s reduced section in function of time is calculated based on the force measurements and the reduced section (derived from the optical images and a poisson ratio of 0.22 and 0.3 for PCL and PLCL, respectively, which were in turn derived from the change in width of the non-degraded samples). Fourth, these intermediate calculations allow us to determine the Young’s modulus at the start of each pull and the yield stress and yield strain, if failure occurred before the end-of-stroke. Fifth, the dimensionless relaxation modulus is calculated based on relaxation behaviour of the tensile modulus during the two holding periods. We described this behaviour using a n-term Prony expansion of the dimensionless relaxation modulus $g_{R,n}\left(t\right)$ [24]: (1)$$g_{R,n}\left(t\right)=1-\sum_{i=1}^{n}g_{i}\left(1-e^{-t/\tau_{i}}\right),$$
with $g_{i}$ and $\tau_{i}$ being fitting parameters. A one-term and two-term expansion were fit on the experimental data (see Figure 4). Ultimately, the two-term expansion was chosen as it approached the observed relaxation best. Lastly, The on-machine deformation results are used to validate the visually derived strain data, taking into account a correction for the unwanted deflection of the 3D-printed PLA connection arms and the specific behaviour of the dogbone shape samples. More details on this procedure are given in Appendix A.

## 3. Results

### 3.1. Visual Observations

During testing, the following general observations about the state of the samples were made; these influence the mechanical characteristics. First, and as shown in Figure 2A,D, samples are stained by the degradation medium, where longer submersion times results in a darker color. However, both overall shape and dimensions did not change during degradation. Second, lightly degraded PCL samples produce high contrast optical data and exhibit a clear yield point as depicted in Figure 2B, while for highly degraded samples the images are less clear and the samples break before plastic deformation occurs (Figure 2C). Third, the translucency of PLCL samples limits the contrast in the optical images, preventing reliable measurement of the change in width of the reduced section. In future experiments employing optical measurements, the quality of the data can be improved by following adaptations: (i) The use of fluorescent or luminescent ink to mark the dots, improving visibility. (ii) Using a wider lens aperture to increase sharpness of images. (iii) Applying a polarizing filter to avoid interference of lighting conditions with the water surface. Fourth, PLCL samples do not exhibit a clear yield point, as demonstrated in the force- and displacement curves in Figure 2E. Further, the failure mode of PLCL samples changes during degradation, where more degraded samples display several cracks that appear in the first (or second) low amplitude loading cycle, leading to premature sample failure (Figure 2F). Stress-strain diagrams derived from the non-degraded force-displacement behavior of PCL and PLCL can be found in Appendix A.

### 3.2. Yield Stress

Although the definition of yielding is clear for perfect elastic materials, the yield point for PCL and PLCL is not as clear, due to viscoelastic effects. This is depicted in Figure 2C–F, where the different yield points considered in the tensile testing are indicated with a star. For PCL, the zone of elastic deformation is clearly demarcated, after which the samples either break (Figure 2C), or show local plastic deformations in the form of necking (Figure 2B). For PLCL, global rather than local plastic deformations occur, which manifests itself as a deviation from linear behaviour in the force displacement graph (see Figure 2E). These yield points are indicated by a star in Figure 2. For both PCL and PLCL we report on the measured yield stresses in Appendix A, with yield stress of non-degraded PCL being 14.5 MPa and of non-degraded PLCL being 9.5 MPa. We visualize the change in yield stress during degradation relative to the values of samples submerged for one day in Figure 3A, and not relative to non-degraded samples. This is done to isolate the effect of degradation from the effect of water ingress in the polymer network, which typically happens at a faster timescale as explained below. For PCL, we observe a decreasing yield stress throughout the degradation protocol. However, for PLCL the observations are more surprising. The yield stress after 1 day is less than half of the non-degraded samples, after which the yield stress recovers slightly until day 20, after which it further declines. We believe that this behaviour can be attributed to the ingress of water in the PLCL polymer matrix, drastically changing the mechanical properties. As all tensile tests, and thus also on the non-degraded samples, were performed with samples submerged in a saline solution at 37 °C, we believe that the ingress of water takes longer than 1 h, being the maximum time the samples are submerged in saline water during tensile testing. Overall, substantial degradation is seen for both materials over the course of 70 days.

### 3.3. Elongation at Break

Figure 3B and Appendix A show the mean absolute values of and change in elongation at break. This property describes the strain at which breaking of the sample occurs. For both materials, the samples during the first week of degradation didn’t break before reaching the maximum stroke of the tensile testing machine, hence the true elongation at break remains unknown but exceeds 539% and 651% for non-degraded PCL and PLCL respectively. These data points are thus omitted from the figure. For both materials, only 1 sample failed before end-of-stroke, visualized with a single data point for PCL and PLCL. After 21 days of degradation, all samples failed before end-of-stroke, and decrease in elongation at break can be seen, limited for PCL and substantial for PLCL.

### 3.4. Young’s Modulus

The Young’s modulus of both materials is characterized at small deformations to eliminate viscoelastic and plastic effects. The modulus is measured for each sample in three instances, being at the beginning of each of the three rises in motor displacement, as depicted in Figure 2. Two methods are used to determine the modulus of elasticity. (i) Through the deformation based on optical data, which is a more direct measurement but also shows a higher uncertainty due to the varying quality of the images. (ii) Through the deformation calculated from the encoder data, which is more consistent but a correction needs to be applied to cancel out the compliance of the clamping structure, which includes approximation errors, as discussed in Appendix A. The results for the modulus of elasticity are presented on Figure 3C,D (a version of these graphs, focusing on the first ten days of degradation can be found in Appendix A) and in Appendix A. These graphs indeed show large uncertainty in the results from the optical data, whereas the encoder data seems more consistent. It can also be clearly seen that the modulus at the first rise is significantly higher than at the subsequent two rises, which can be contributed to the viscoelastic relaxation after the first rise. Surprisingly, no significant change in elastic modulus throughout the degradation can be seen for PCL (Figure 3C). For PLCL on the other hand, the modulus decreases after 42 days of degradation.

### 3.5. Viscoelastic Relaxation Modulus

The effect of degradation on the viscoelastic behaviour of the polymers is identified by means of changing viscoelastic coefficients in the Prony expansion of the relaxation modulus (Equation (Equation 1)). Figure 4 shows the relaxation of the dimensionless tensile modulus $E/E_{0}$ (where $E_{0}$ is the modulus at the onset of the stress relaxation) of non-degraded PCL (A) and non-degraded PLCL (B) when the displacement and thus material strain remains constant. A one-term and two-term Prony expansion, corresponding with the Maxwell and the Maxwell–Wiechert model for viscoelasticity respectively [22], were fitted to these relaxation curves. The two-term Prony expansion of the dimensionless relaxation modulus $g_{R,2}\left(t\right)$, according to Equation (Equation 1) seems to correspond best to the relaxation, resulting in values for the coefficients $g_{1}$ and $g_{2}$, which represent a fraction of the dimensionless relaxation modulus and $\tau_{1}$ and $\tau_{2}$, which represent the time constants associated with these fractions. At day 0, the values of these coefficients are for PCL: $g_{1}=0.11,\;\tau_{1}=0.89\;s,\;g_{2}=0.18,\;\tau_{2}=9.05\;s$; and for PLCL: $g_{1}=0.16,\;\tau_{1}=2.00\;s,\;$$g_{2}=0.34,\;\tau_{2}=25.14\;s$. Relative to PCL, PLCL thus shows a much slower response to dynamic loading, and a longer relaxation period should have been provided for more accurate measurement of the viscoelastic relaxation of PLCL. The change of these coefficients during degradation is visually depicted on Figure 3E for PCL and F for PLCL, and can also be found in Appendix A. Again, a substantial change in the coefficients with respect to non-degraded PLCL can be seen, which is a result of the ingression of water. Therefore, we display for both polymers the relative change in coefficients with respect to day 1. Further, the viscooelastic coefficients remain fairly constant throughout the degradation. The large uncertainty in some data points can be attributed to the fitting of Equation (Equation 1) to noisy data, but does not undermine the general trend. After 70 days of submersion in the degradation fluids, only one sample of PCL remained intact during the first hold, hence no standard deviation is displayed. Starting from 56 days of degradation, the PLCL samples started tearing before the first hold was finished, so no viscoelastic behavior could be quantified.

## 4. Discussion

The previous paragraph presented the yield stress, elongation at break, viscoelastic relaxation parameters and the modulus of elasticity for small deformations of both PCL and PLCL, both before and after degradation in small intestinal fluids. These results can serve as a practical aid in designing PCL and PLCL implants, when simulating their deployment and behavior under mechanical loading. However, the question arises on how these results compare to literature.

Considering the non-degraded materials, Table 1 compares the mechanical properties obtained from this study to previously established values for the yield stress, elongation at break and elasticity modulus of PCL and PLCL. Despite the possible influence of material origin, handling and processing to these properties, the results of our tensile tests are comparable to previous studies. The range of elastic moduli reported here (PCL: 203–554 MPa, PLCL: 21–51 MPa) are to be interpreted as follows: due to the viscoelastic behavior of both materials, the materials will behave with a higher stiffness during the first loading step. After relaxation of the initial load, the approximate modulus of elasticity will settle at a lower value (visible in Figure 3C,D). This has implications on the use of the reported mechanical properties in the design of biodegradable implants, as these often experience a substantial initial load during implantation and deployment, followed by more regular loading cycles. For some applications, the materials’ viscoelastic behavior might be important to the implant design, however, no previous study reported viscoelastic relaxation data that is readily useful in a simulation environment. A remarkable difference can be observed when comparing the relaxation of non-degraded PCL and PLCL (see Figure 4). The stress in the PCL sample is reduced by roughly 13% while the stress reduction in PLCL is 30%, and is sustained over a longer time. This difference can be explained by the semi-crystalline nature of PCL, while PLCL exhibits microphase separations due to the presence of the l-Lactide units [25]. Appendix A lists the coefficients of the Prony expansion of the dimensionless relaxation modulus (Equation (Equation 1)) and these can readily be used in finite element software packages for both PCL and PLCL. Such a simulation has been performed using Abaqus (Simulia) finite element software on a dogbone-shaped sample, where the stress–relaxation curve is simulated and validated using experimental data (Appendix A for details and results.).

Considering the change of mechanical properties during degradation, no studies have been found that use small intestinal fluids as degradation media. However, other media have been explored in literature. Karjalainen et al. [20] measured the change in yield stress and modulus of elasticity of PCL in a phosphate buffer solution (PBS) at 37 °C over the course of 10 weeks and reported quasi unchanged properties for PCL. Chang et al. [18] investigated the degradation of PCL films in various digestive fluids. They reported an increased weight loss of the films when degraded in saliva and gastric juice, however, a decreased weight loss in bile. This suggested a non-negligible influence of the specific degradation environment, and more specifically of the alkalinity and the presence of various digestive enzymes. The study by Dias et al. [17] on the degradation of electrospun PCL fiber meshes indicated a difference in modulus of elasticity and elongation at break when the meshes had been degraded by PBS or PBS with added lipase. It is thus clear that the specific degradation environment of small intestinal fluids could influence the change in mechanical properties of PCL and PLCL. Indeed, as Figure 5A indicates by comparing the changing yield stress and elasticity modulus of PCL in both PBS (as reported by Karjalainen [20]) and in small intestinal fluids, the yield stress measured after degradation in these fluids decreases faster than in PBS. However, the change in modulus of elasticity is comparable between the two degradation fluids. This is confirmed by a comparison with a much broader spectrum of elastic moduli reported in a review by Bartnikowski et al., shown in Figure 5B. This raises further questions on the specific degradation mechanisms in small intestinal fluids and the indivual influence of the various microbes, enzymes and bile acids present in this environment, which could not be answered with the data gathered during this study. No previous degradation study of poly-l-lactide-*co*-caprolactone with copolymer ratio of 70% l-Lactide and 30% Caprolactone could be found to compare the data from this study with. However, comparing the degradation behavior of the tested PCL and PLCL samples (Figure 2 and Figure 3) provides insight in the interaction between the degradation environment and the chemical nature of the materials. The premature failure of PLCL can be attributed to the presence of the l-Lactide units. Surrounded by the more hydrophobic Caprolactone chains, they are more prone to hydrolysis, increasing the rate of the polymer chain scission [25]. In contrast, PCL is a hydrophobic semi-crystalline polymer, exhibiting a high resistance to hydrolysis. Enzymatic degradation is expected to accelerate the cleavage of the polymer chains, that begins in the amorphous regions of the polymer, initially increasing its crystallinity [18].

## 5. Conclusions

This study reports the yield stress, elongation at break, viscoelastic relaxation parameters and approximate modulus of elasticity at low deformation of PCL and PLCL, both in non-degraded state and after degradation of up to 10 weeks in fresh human small intestinal fluids at 37 °C. Whereas previous studies focused on the quasi-static mechanical properties of pristine samples of PCL and PLCL, this work also reports on their dynamic properties by means of the viscoelastic Prony expansion coefficients. Non-degraded PCL samples showed a yield stress of 14.5 MPa, elongation at break of more than 539% and Young’s modulus of 203–554 MPa, while non-degraded PLCL showed a yield stress of 9.5 MPa, elongation at break of more than 651% and Young’s modulus of 1.3–3.1 MPa. Degradation of the materials in small intestinal fluids caused a substantial decrease in yield stress and elongation at break of both PCL and PLCL, while the elastic modulus of PCL and viscoelastic coefficients of both materials remained unchanged, and the modulus of PLCL only slightly decreased after 6 weeks of submersion. Further, water ingress into the polymer network substantially changes both the absolute yield stress, elongation at break, Young’s modulus and the viscoelastic properties of PLCL. It is therefore recommended to test PCLC after submerging it in water for one day. This allows the polymer network to fully saturate with water without showing signs of degradation. A similar effect in PCL has not been observed. These results show the potential of PCL and PLCL for use in biodegradable implants in the small intestine, and enable the designer to better estimate and simulate the implant’s behavior and specific lifetime when subjected to deployment and regular mechanical loading by providing time-dependent mechanical properties that can be directly implemented in FEA simulation tools.

This study focuses on the mechanical effect of degradation of PCL and PLCL, however, a complete understanding of the chemical processes behind the degradation falls outside the scope. Follow-up studies focusing on mass loss, change in crystallinity and molecular weight throughout the degradation in small intestinal fluids, would give more insights in the underlying mechanisms. Additionally, a thorough fracture analysis by means of stereo microscopy or scanning electron microscopy would provide new insights into the failure mechanisms associated with degraded PCL and PLCL stretching.

## Figures and Tables

**Figure 1 polymers-15-02964-f001:**
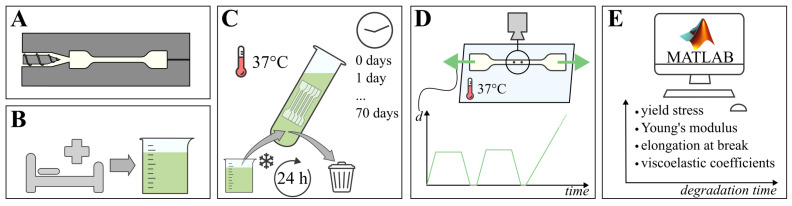
Schematic of the material testing procedure; (**A**) tensile testing samples of PCL and PLCL are injection molded, (**B**) fresh small intestinal fluids are collected, (**C**) samples are submerged for their respective degradation time at 37 °C, while fluids are refreshed every 24 h, (**D**) tensile testing by imposing displacement on samples while recording reaction force and visual data, (**E**) post-processing of recorded data using Matlab, resulting in mechanical material properties over degradation time.

**Figure 2 polymers-15-02964-f002:**
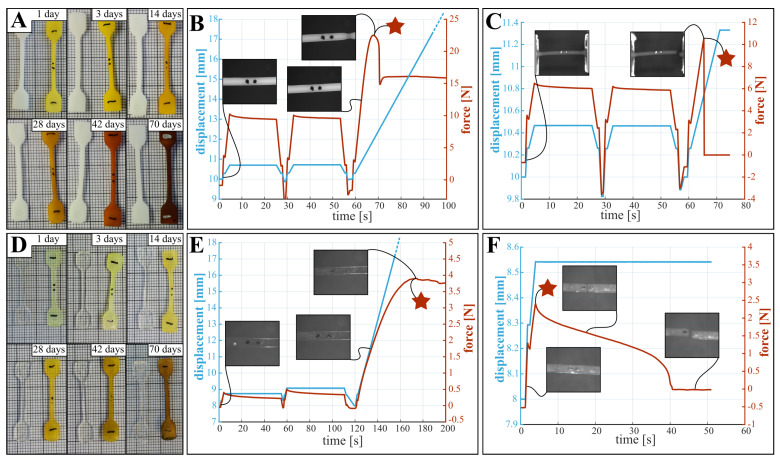
Observations made after degradation and during testing for PCL (**top panes**) and PLCL (**bottom panes**). The graphs show typical displacement- and force curves during tensile testing. (**A**,**D**) samples exhibited staining throughout degradation, without dimensional change. (**B**) Non-degraded PCL samples produce sharp high contrast images, and a well defined yield point as the samples show necking. (**C**) More degraded PCL samples (here 70 days) produced lower contrast images and display brittle fracture. (**E**) Non-degraded PLCL samples are partially transparent, producing low contrast images, without a clear yield point. (**F**) More degraded PLCL samples (here 56 days) display premature fracture, due to the appearance of multiple cracks in the reduced section. The locations of the considered yield points are indicated with a star.

**Figure 3 polymers-15-02964-f003:**
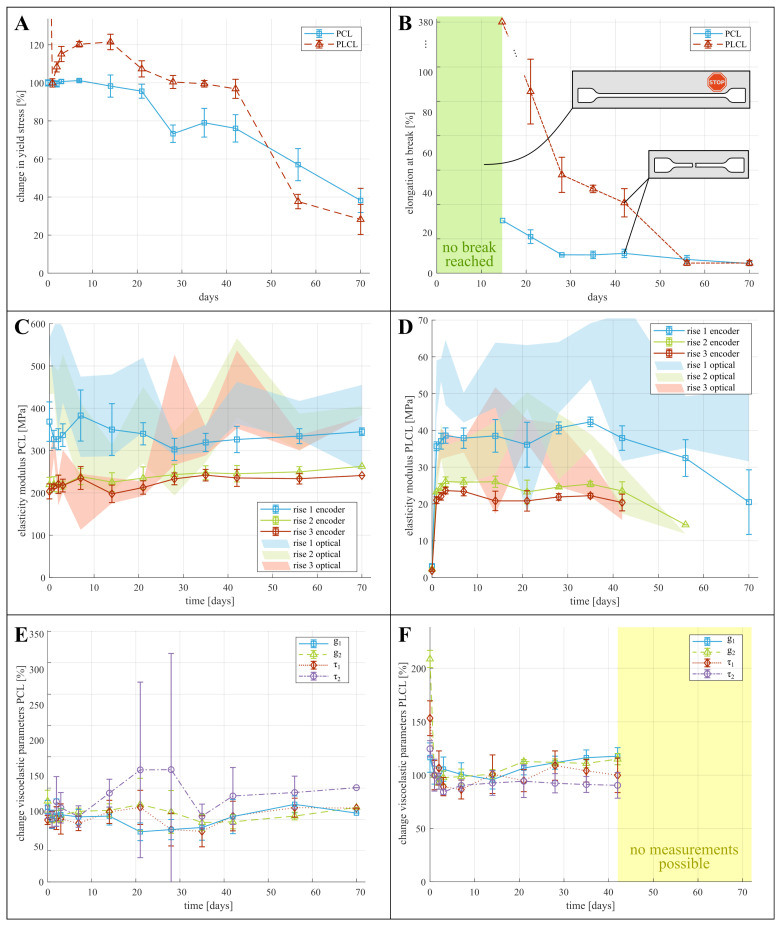
Graphs displaying change in mechanical properties of PCL and PLCL. (**A**,**B**) change in yield stress and elongation at break relative to initial value, (**C**,**D**) absolute values of elasticity modulus, based on optical data (shaded area) and based on encoder data (solid lines). (**E**,**F**) change of each viscoelastic parameter (see Equation (Equation 1)) relative to their initial, non-degraded, values.

**Figure 4 polymers-15-02964-f004:**
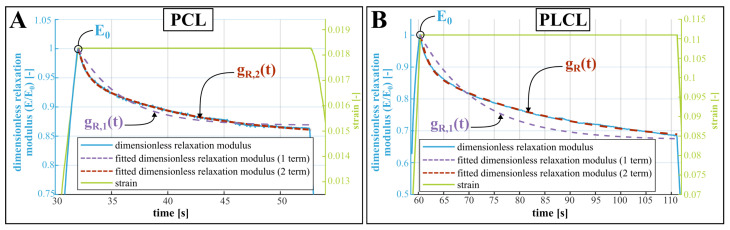
Viscoelastic relaxation of a non-degraded PCL (**A**) and PLCL (**B**) sample, showing the dimensionless tensile modulus $E/E_{0}$, where $E_{0}$ is the modulus at the onset of the stress relaxation. While the strain remains constant, the modulus decreases. This exponential decrease is fitted by the dimensionless relaxation modulus $g_{R,n}\left(t\right)$ (Equation (Equation 1), using one term and two terms for comparison).

**Figure 5 polymers-15-02964-f005:**
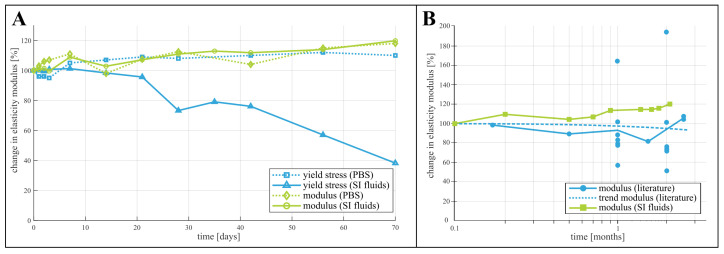
Comparison of the results with data from previous studies. (**A**) rate of change of yield stress and elasticity modulus of PCL in buffer solution (PBS) [20] versus small intestinal fluids and (**B**) comparison of data of the degradation of the elastic modulus from a literature review [3] and our results.

**Table 1 polymers-15-02964-t001:** Comparison of non-degraded PCL and PLCL mechanical properties.

	Yield Stress [MPa]	Elongation at Break [%]	Modulus [MPa]	Source
PCL	16	>100	260	Hiljanen-Vaino [14]
	-	700–900	200–400	McMahon [2]
	16	80	400	Engelberg [15]
	17.82	>438	440	Ragaert [13]
	8.2–17.8	80–800	251.9–440	Bartnikowski [3]
	14.5	>539	203–554	this work
PLCL	-	400	12–128	Fernández [9]
	9.5	>651	21–50.8	this work

## Data Availability

The data presented in this study are openly available in an open access KU Leuven repository (RDR) (doi:10.48804/WHR9UM).

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
