# Peer review of "Effect of Degradation in Small Intestinal Fluids on Mechanical Properties of Polycaprolactone and Poly-l-lactide-co-caprolactone"

_polymers, 2023, doi:10.3390/polym15132964_

Round 1

Reviewer 1 Report

In the manuscript by Peerlinck et al., the authors presented a study of the degradation behaviors of polycaprolactone and poly-L-lactide-co-caprolactone in a human-mimic small intestinal environment. The authors designed a full set of automation systems to perform time-dependent degradation experiments, monitoring the mechanical performance evolution of PCL and PLCL under these conditions. The manuscript is well-written and is a great piece of joy to read. Additionally, this reviewer particularly likes the way the authors put up the structures of the manuscript, with Figure 1 extremely helpful to understand what the authors were trying to convey from a high level. Overall, this manuscript is of high scholarly quality with most of the experiments performed solid and sound. Hence, this reviewer would like to recommend the publication of this manuscript with the following general/technical comments for the authors to consider.

(1)   The focus of the manuscript is the degradation behavior of PCL and PLCL in a small intestinal mimic environment. The mechanical characterizations of both materials were comprehensive but what the authors did not elaborate on was the chemical nature of the materials and the small intestinal environment. The small intestine is a harsh digestive environment full of various microbes, enzymes, and bile acids. Examining the chemical composition of PCL and PLCL, one might find it easier to understand why PLCL is more susceptible to degradation in small intestinal environments. This vulnerability stems from the fact that the lactide unit in PLCL is more prone to hydrolysis due to its hydrophobic caprolactone. Therefore, it would be beneficial if the authors could discuss more on the chemistry of degradation behaviors of both materials.

(2)   A similar issue arises with the viscoelastic behaviors of PCL and PLCL. The authors only observed viscoelastic properties in PLCL but not in PCL. This is because PCL is a semi-crystalline polymer and has similar mechanical properties as polyethylene and polypropylene. However, PLCL has microphase separations due to the presence of lactide units (refer to the review here, J. Appl. Polym. Sci. 2022, 139, e53091). It would be helpful to have these discussions rather than from a pure mechanical engineering point of view.

(3)   Some nitpicking formatting suggestions: the “L” in the L-lactide needs to be italicized. The copolymer notation “co” in “poly-L-lactide-co-caprolactone” needs to be italicized.

(4)   Some comments related to the mechanical testing method:

(a)   For the tensile tests performed in a submerged saline solution, did the authors consider the buoyancy from the grippers and the sample specimens and subtract it? Providing a blank (without sample specimens) curve could help confirm that the buoyancy's contribution is not significant.

(b)   This reviewer understands that the authors were attempting to provide an optical measurement to estimate the actual stress in the sample. However, it is evident that this measurement is less accurate and prone to considerable fluctuation. The authors should consider methods to improve the accuracy of the optical measurement.

(c)   The relaxation was only fitted with a Maxwell-Wiechert model, and the authors should consider fitting the data with the Maxwell model to show the difference between the two and see if a two-term Maxwell-Wiechert model is necessary to understand the viscoelastic behaviors. Additionally, the fitting time frame was only 15 seconds, which is not sufficient to give reliable results. It would be better to collect data for a longer period. (See a relevant paper for reference: Nat. Mater. 2016, 15, 326–334.)

Reviewer 2 Report

Dear Authors,

Your research is impressive and thoroughly performed, but I somewhat feel it fails to completely conform to the purpose. I state this because somewhere in the experimental part it felt like reading a research on method validation and I completely forgot the true aim of your reserach. I do not challenge your method, it is well presented and supported with results and conclusions, but the main focus should be on the alteration of mechanical properties and behaviour of your tested samples. 

I advise you to include the classical stress - strain diagrams, which would ease the understanding for most readers that are unfamiliar with mechanical testing. Also a fracture analysis, when possible, even at low magnifications, using a stereo microscope, would enhance the understanding of the mechanical behavior, failure mechanism and should provide valuable insight on the degradation influence.

Also the charts comparing the values of the mechanical parameters show a clutter of data in the first 10 days, that, unfortunately, is impossible to analyze. I recommend using a logarithmic axis for Ox, perhaps this format would enhance the perceptiom. If this does not work, you could introduce an inset depicting the region 0-10 days.

There are several slips in English language and style that I have marked in the attached document. Revision is advised.

In line 70 you state "design of experiment", a statement that implies another methodology, I would advise experimental design/experimental protocol or another terminology that would not create confusion.

The tensile testing was performed on a Zwick Roell testing machine, it should be tested accordingly, in line 118 the way it is stated it implies that tensile testing is performed using a ZwickRoell protocol - for those unfamiliar with mechanical tests. I also advise stating the standard used as guide, even if the protocol is proprietary.

There are some lines that need rephrasing, sunch as lines 169-170, fracture failing caused by crack propagation that already appears ... - I suggest rephrasing.

I respect the effort in this research, and my main request is adressing the chart presentation.

My best regards.

English language and style are fine, several corrections are required, see the attached document.
